**Data Availability Statement:** All relevant data are within the manuscript and its Supporting Information files.

# Drug-resistant tuberculosis care and treatment outcomes over the last 15 years in Ethiopia: Results from a mixed-method review of trends

E. Tesema[1]*, M. Biru[1], T. Leta[2], A. Kumsa[2], A. Liaulseged[2], G. Gizatie[1], T. Bogale[1], M. Million[1], D. G. Datiko[3], A. Gebreyohannes[1], Y. Molla[3], N. Hiruy[3], M. Mebnga[4], P. G. Suarez[5], Z. G. Dememew[3], D. Jerene[6]

**1** USAID Eliminate TB Project, KNCV Tuberculosis Foundation, Addis Ababa, Ethiopia, **2** Ministry of Health, National TB, Leprosy and other Lung Disease Program, Addis Ababa, Ethiopia, **3** USAID Eliminate TB Project, Management Sciences for Health, Addis Ababa, Ethiopia, **4** KNCV Tuberculosis Foundation, Moscow, Russia, **5** Management Sciences for Health, Arlington, VA, United States of America, **6** KNCV Tuberculosis Foundation, The Hague, The Netherlands

* emawayish.tesema@kncvtbc.org

## Abstract

### Background and objectives

Drug resistant tuberculosis (DR-TB) remains a global challenge with about a third of the cases are not detected. With the recent advances in the diagnosis and treatment follow-up of DR-TB, there have been improvements with treatment success rates. However, there is limited evidence on the successful models of care that have consistently registered good outcomes. Our aim was to assess Ethiopia's experience in scaling up an ambulatory, decentralized model of care while managing multiple regimen transition processes and external shocks.

### Methods

This was a cross-sectional, mixed-method study. For the quantitative data, we reviewed routine surveillance data for the period 2009–2022 and collected additional data from publicly available reports. We then analyzed the data descriptively. Qualitative data were collected from program reports, quarterly presentations, minutes of technical working group meetings, and clinical review committee reports and analyzed thematically.

### Results

The number of DR-TB treatment initiating centers increased from 1 to 67, and enrollment increased from 88 in 2010 to 741 in 2019, but declined to 518 in 2022. A treatment success rate (TSR) of over 70% was sustained. The decentralized and ambulatory service delivery remained the core service delivery model. The country successfully navigated multiple regimen transitions, including the recently introduced six-month short oral regimen. Several challenges remain, including the lack of strong and sustainable specimen transportation

**Funding:** The Global Health Bureau, Office of Health, Infectious Disease, USAID financially supported the study through USAID Eliminate TB Project under the terms of Agreement No. 72066320CA00009. The funding was provided for Management Sciences for Health (MSH) organization for implementing USAID Eliminate TB Project. The authors' views expressed in this publication do not necessarily reflect the views of USAID or the US Government.

**Competing interests:** the authors have declared that no competing interests exist.

system, lack of established systems for timely tracing and linking of missed DR-TB cases, and data quality issues.

## Conclusions

Ethiopia scaled up a decentralized ambulatory model of care, kept up to date with recent developments in treatment regimens, and maintained a high TSR, despite the influence of multiple external challenges. The recent decline in case notification requires a deeper look into the underlying reasons. The feasibility of fully integrating DR-TB treatment and follow up at community level should be explored further.

## Introduction

Tuberculosis (TB) remained a major public health problem worldwide, affecting over 10 million people each year [1]. About 3–4% of newly diagnosed TB patients have drug resistant forms of TB, and the rate is 18–21% among persons with previous history of treatment [2]. Following the Corona virus disease 2019 (COVID-19) pandemic, the number of people treated for DR-TB in 2020 dropped by 15%, compared to 2019 estimates of half a million cases [3]. Despite a decline in DR-TB patients enrolled in treatment, there has been progressive improvement in treatment outcomes over the years where treatment success rate reached 60% in 2022. However, this translates to 40% unfavorable outcomes calling for better strategies to improve treatment outcomes in high burden settings [2].

The recently recommended all-oral short treatment regimen is one of the major strategies in improving the management of DR-TB, shortening the treatment duration and improving adherence and treatment outcome [4]. However, there are concerns about acquiring resistance with the widespread implementation of regimens containing bedaquiline (BDQ), which require close follow-up of the program and documenting lessons from early adopter countries [5, 6].

Ethiopia is among a high TB and TB/HIV burden countries who have registered consistent decline in TB incidence, by an average of 5–10%, from 192 in 2015 to 119 per 100,000 in 2021. However, there was 7% increase to 126/100,000 which was believed to be due to multiple external shocks including conflict, COVID-19 and natural disasters [1]. Despite these challenges, the incidence and mortality rates of TB declined by 31.1% and 34.6%, respectively, compared with the 2015 baseline. As a result, Ethiopia attained the 2020 End TB milestones [7]. The latest drug resistance survey (DRS) report revealed that 1.1% of new and 7.5% of previously treated TB cases were estimated to have MDR-TB in 2019 [8].This is a significant decline from the 2005 DRS rates of 2.7% and 14% among new and previously treated cases, respectively [9]. Based on the updated estimates for DR-TB burden, Ethiopia is no longer among the 30 high MDR/RR-TB burden countries [8].

We previously reported successes and challenges associated with Ethiopia's scaled-up implementation of the ambulatory model of care [7]. Since then, the country has continued implementing the World Health Organization's (WHO) updated recommendations, which are based on the best available evidence on model of care and person-centered approaches. Since 2016, there have been various regimen transitions adopted by the country for programmatic implementation. These transitions happened despite multiple challenges, including COVID-19 and widespread conflict with consequent massive internal displacement which could have a negative impact on the overall TB/DR-TB-related program implementation [10, 11].

Our aim was to assess Ethiopia's experience in scaling up an ambulatory, decentralized model of care with a good treatment success rate (TSR) while managing multiple regimen transition processes and external shocks. The findings will contribute to further refinement of implementation approaches and policy guidance in the country and beyond.

## Methods

### Study setting and design

Ethiopia is a low-income country with per capita income of 1028 USD. The country's population, which is predominantly rural, is estimated to exceed 123 million and the average life expectancy is 65 years. Table 1 summarizes key socio-demographic and health indicators for Ethiopia. The country has a three-tiered pyramidal health system care system with primary health care facilities forming the broad base, secondary health care facilities in the middle, and tertiary care at the apex (see figure as supplemental information). Ethiopia follows a decentralized service delivery model as much as possible with the bulk of the service provided at primary care level. For DR-TB care, the country followed the healthcare tier system in setting up the services. Treatment Initiating Centers (TICs), where patients initiate their treatment and temporarily admitted if needed were initially set up at tertiary level but later much of the TIC role moved to secondary level. Treatment follow-up centers (TFCs) are facilities where patients receive their daily treatment after they are transferred from the TICs. Each TIC has a designated number of TFC within their catchment areas, defined by administrative regions and zones. Detailed description of the roles and responsibilities of TICs and TFCs is provided [7].

A cross-sectional programmatic data review was done for the period 2009–2022. The assessment followed a mixed study design and used both quantitative and qualitative data collection approaches.

### Data sources

Publicly available and national TB program reports were used. Programmatic information from treatment initiation centers (TICs), treatment follow-up centers (TFCs), activity reports, and meeting reports were reviewed to complement the quantitative data. All MDR/RR-TB patients reported in the WHO annual global tuberculosis reports for 2009–2022 were reviewed in this study.

### Data collection and procedure

Routine service data were collected from the national Health Management Information System (HMIS) and WHO annual reports for 2009–2022 to document and synthesize the trend of MDR/RR-TB enrollment and treatment success. Data were collected from January 1 to

**Table 1. Key socio-economic and health indicators.**

|  | Value for Ethiopia Characteristic |
| --- | --- |
| Estimated population (millions)* | 123,379,924 |
| Estimated annual per capita income (US$) * | 7510 |
| Life expectancy at birth* | 65 years |
| Infant mortality rate* | 34 |
| % of government expenditure on health* | 28.3% |
| % TB patients suffering catastrophic cost [12] | 52% |

Source*: The World Bank data, 2022

February 28, 2023. It was collected using a tailored case report format from the District Health Information System-2 (DHIS2) and/or standard of care (SOC) tool. The SOC tool is a supervision tool that monitors the implementation of DR-TB service at TICs and TFCs. In addition, national TB program reports and TB and Leprosy (TBL) annual bulletins were reviewed. Trends of MDR/RR-TB enrollment and composite final treatment success of MDR/RR-TB cases were computed.

The quantitative data were supplemented with qualitative programmatic data, including reports, quarterly presentations, minutes of technical working group meetings, and clinical review committee reports.

## Data entry and analysis

Quantitative data were entered and analyzed using Excel. Descriptive statistics were computed to get summary results. We conducted a manual thematic analysis of qualitative data. We have synthesized those findings in connection to MDR/RR-TB patient care and services in three thematic areas: *Growing demand for MDR/RR-TB service access*, *MDR/RR-TB Service expansion and data quality*, *and Resource limitation and lack of locally evidenced solutions*. The descriptions of these themes are embedded in the results and discussions.

## Ethical consideration

The study carries no more than minimal risk. Hence, permission to use the routinely collected health facility data on tuberculosis services for program implementation was requested from Oromia Health Bureau in Addis Ababa, Ethiopia to proceed and conduct the data analysis. Routinely collected data form health facilities who are implementing TB program services was analyzed for service improvement, provide recommendation, documentation, and possible publication for others to learn the good practice. All potential identifiers were removed from the format to ensure anonymity.

## Results

### Inception of the MDR-TB program

Ethiopia conducted the first national DRS in 2005 and found a DR-TB rate of 2.3% among new and 17% among previously treated TB cases. This was followed by a Green Lights Committee-approved pilot program to treat MDR/RR-TB patients in 2009 at St. Peter's Specialized Hospital in Addis Ababa. The program was successful in terms of patient enrollment, good case holding, and improved the treatment outcome, which led to the National TB Program (NTP) scaling up the service and establishing the national Programmatic Management of DR-TB (PMDT) technical working group to support the program.

### Hospitalized model of care (2009–2011)

Between 2009 and 2011, the NTP introduced a hospitalized model of DR-TB care in the first two treatment centers during the initial two years of DR-TB care program initiation. In the hospitalized model of care, the facility provides patient care until the patient has culture and smear conversion. During this period, 208 DR-TB patients were enrolled to treatment according to the WHO annual global TB report 2010/11.

### Clinic-based ambulatory model of care (2012–2022)

During this period, national policies and guidelines were developed, monitoring and evaluation system established, additional DR-TB treatment facilities renovated, access to drug

susceptibility testing (DST) improved, and capacity of health care workers built, all of which further improved DR-TB case finding; enrollment of patients on available, effective treatment; admission capacity of health facilities; and well-coordinated programs. Additional TB culture and DST centers were established in four regions to improve the country's diagnostic capacity. Health facilities were networked with nearby culture and DST laboratory services. During the same period, 10 additional MDR-TB treatment centers were opened.

This period has brought a major shift in the program design and treatment service delivery model from hospitalized care to a clinic-based ambulatory model of care. In addition, the first-of-its-kind national implementation guide and protocol were prepared for implementation of an ambulatory treatment model at service delivery centers. This supported the rapid decentralization of PMDT services in the local context. The focus gradually shifted to the introduction of rapid molecular diagnostics (X-pert MTB/RIF Assay), introduction of new WHO-endorsed treatment regimens, and expansion of treatment centers to more public hospitals in all regions. The number of TICs increased from 1 in 2009 to 67 in 2022. Between 2014 and 2015 alone, 42 TICs were opened with more than 700 satellite treatment follow-up centers linked to them (Fig 1).

**The expansion was driven by "growing demand for service access".** The growing demand for managing MDR-TB patients in the country forced the Ministry of Health to come up with a mixed model, with inpatient care for treatment initiation and ambulatory care for treatment follow-up. This approach was found to improve case finding and enrollment and accelerate expansion of DR TB services at the regional level.

The quality of the established approach has been ensured through strong coordination, joint supportive supervision, clinical mentorship, and catchment area meetings. Other activities include involving TICs and TFCs in clinical case consultation in collaboration with specialists and subspecialists to manage difficult and complicated DR-TB cases.

**Regimen transition process.** From December 2009 to 2016, almost all newly diagnosed MDR/RR-TB patients received a standardized longer regimen with an intensive phase

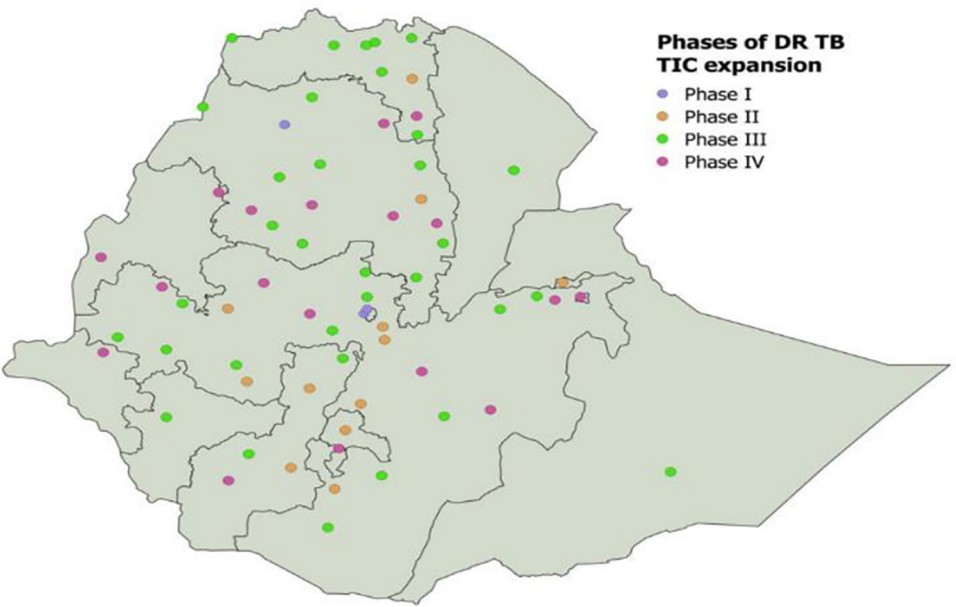

**Fig 1. Phased site expansion by calendar period.** It is a map that shows the phase expansion by calendar period (Phase I: 2009–2011; Phase II: 2012–2013; Phase III: 2014–2015; Phase IV: 2016–2022).

comprising pyrazinamide, capreomycin, levofloxacin, prothionamide (ethionamide), and cycloserine for 8 months (8 Z-Cm6-Lfx–Pto (Eto)–Cs) and a continuation phase for 12 months with all oral drugs but not the injectable capreomycin (12Z-Lfx–Pto (Eto)–Cs). Depending on the patient's history of using the drugs and isolate susceptibility, patients were put on individualized regimens [12, 13].

In 2016, based on the WHO interim guidance document, patients were started on BDQ and delamanid (DLM) based individualized regimen if they were eligible [14, 15]. Since 2017, Ethiopia has implemented a standardized all-oral BDQ-based long treatment regimen. A shorter treatment regimen with injectables lasting 9 to 12 months was implemented as the standard regimen nationwide from April 2018 to February 2021 for newly enrolled MDR/RR-TB patients in addition to the longer regimen. The shorter regimen comprised four to six months of Km (Am)-Mfx-Pto(Eto)-Cfz-Z-Hhigh-dose-E and five months of Mfx-Cfz-Z-E. Another novel regimen—an all-oral BDQ shorter treatment regimen, which has oral BDQ in place of the injectable—was rolled out in Ethiopia in February 2021 (Fig 2).

**Enrollment of patients with TIC expansion and improved treatment success.** DR-TB patient enrollment to second-line drugs increased with the expansion of the TICs. Each expansion phase was linked to flagship TB projects that facilitated changes in the models of care. For example, the shift to the ambulatory model of care, supported by the Help Ethiopia Address Low TB Performance (HEAL TB) project, contributed to an increase in patients' enrollment.

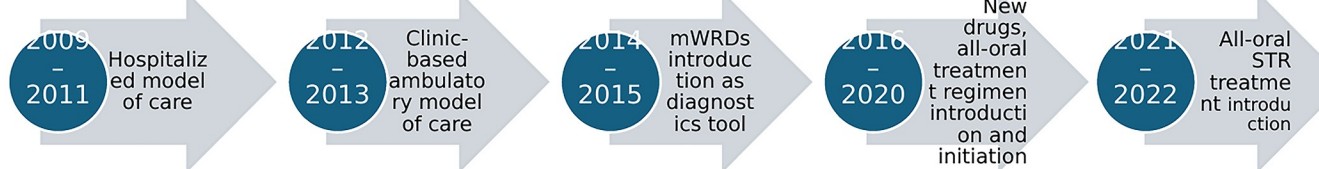

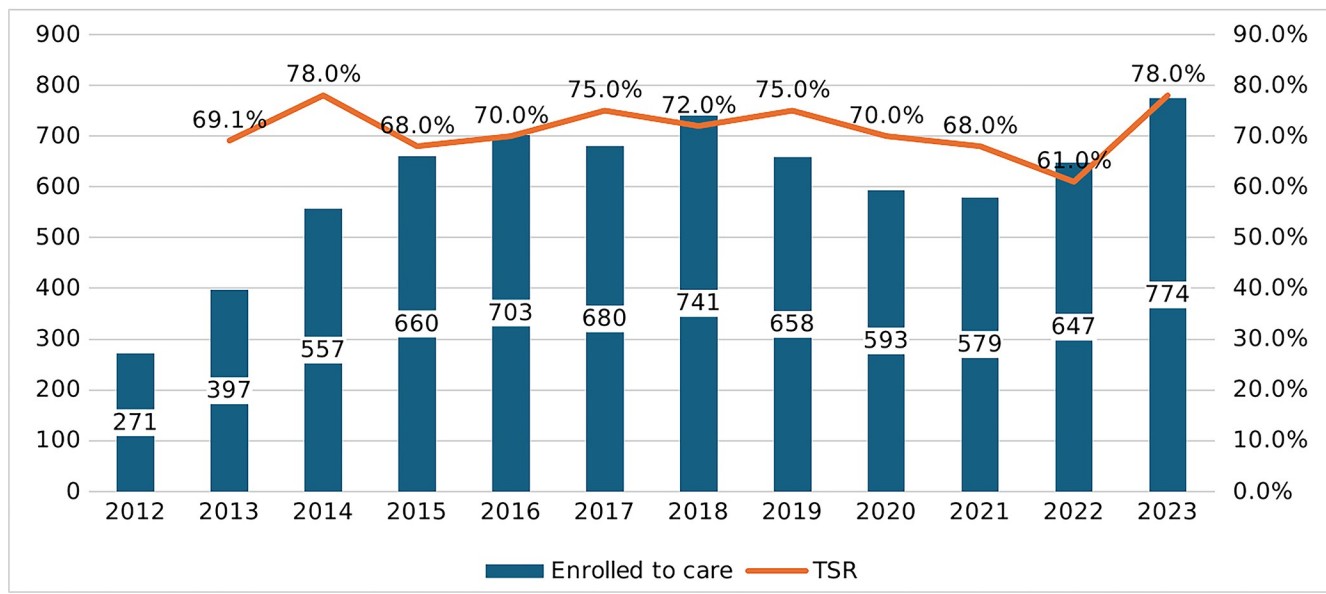

**Fig 2. Timeline of regimen transition, model of care (2009–2022) with number of patients enrolled and final treatment outcomes (2012–2022).** It shows DR TB treatment transition period: prior to 2016 was injectable based long treatment regimen; 2017 and beyond was regimens with newer drugs; 2018 beyond were short treatment introduction.

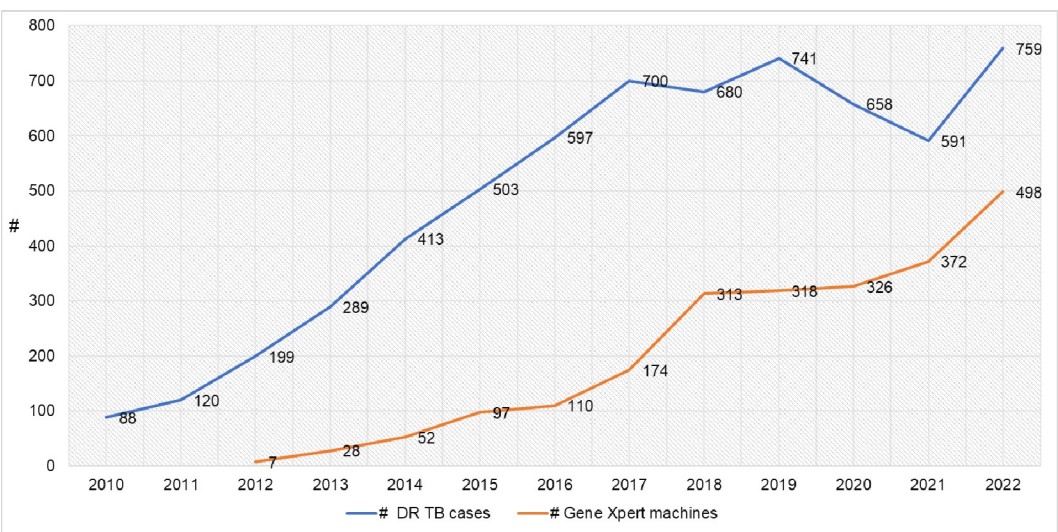

**Fig 3. DR TB Enrollment and GeneXpert scale up trend, 2009–2021 (WHO Global reports, 2010–2022).** It shows DR TB Enrollment and GeneXpert scale up trend from 2009–2021.

(Fig 2). In general, the DR TB case detection has been correlated with the scale up of the GeneXpert machine in the country (Pearson Correlation = 0.82, p-value = 0.002) (Fig 3).

The national DR-TB treatment coverage rate steadily improved and reached 43% in 2022 (national annual report). The national patient retention to end of treatment with successful treatment outcome (>70%) is one of the highest among high DR-TB burden countries (Fig 2).

As of early 2020, Ethiopia is no longer a high DR-TB burden country. The annual national report of treatment outcomes is more than the average global reports of treatment success for both long and short treatment regimens. (Fig 4).

**Challenges encountered during program implementation.** Several challenges were encountered during the implementation of the project. Security issues in some regions of the country hampered access to services. Addressing the need for the pastoralist community is another challenge due to the nature of the community lifestyle.

> *"Some health centers were not providing AFB microscopic in pastoralist communities, because of their way of life, and limited access to health facilities".—HEAL TB Project 2011–2016 final report*

Among the strategies to address the pastoralist community was deploying a mobile van in the community. Although vans have been deployed, their utilization was not as expected. More time was needed to adapt to a new way of working.

The existing approach of using postal services as means of sample transportation to GeneXpert sites is not adequate or efficient, and the lack of a strong and sustainable specimen transportation system is a significant challenge; another is the lack of an established system for timely tracing and linking of missed TB/DR-TB cases. Ensuring data quality is an ongoing challenge for the country.

> *"The evidence shows that, the data collected by the HMIS requires further quality check on regular basis."*

TBL Annual Bulletin No 7, 2021, Ethiopia

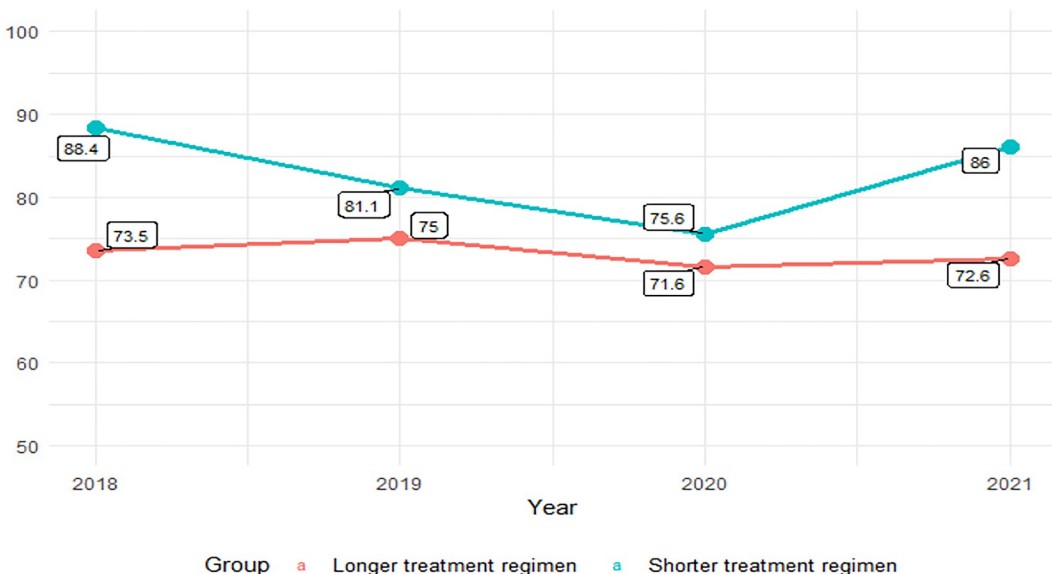

# Outcomes for cohorts of people treated for MDR/RR-TB are reported two year later than the reporting year

**Fig 4. Treatment success rate trend of MDR/RR-TB cases disaggregated by regimen in Ethiopia# 2018–2021 (source: Annual national TB program report).** It shows treatment success rate trend of MDR/RR-TB cases disaggregated by regimen in Ethiopia# 2018–2021. # Outcomes for cohorts of people treated for MDR/RR-TB are reported two year later than the reporting year.

## Discussion

The review of Ethiopia's DR-TB program showed significant improvement in treatment access between 2011 and 2021, as defined by the number of TICs expanding from just one at baseline to 67 by the end of the analysis. At the same time, the country successfully managed the transition process of multiple regimens that were optimally implemented throughout the country.

The model of care transition from mandatory hospital-based treatment of patients to an ambulatory model with limited admission of patients has had a significant impact on improving access to MDR/RR-TB treatment and care services for all patients on waiting lists in different regions of the country and clearing the backlog.

Experiences from other countries have also demonstrated the advantage of service decentralization. In South Africa, services were decentralized up to the community level, enabling site expansion from 17 to 658 sites and increasing the treatment success rate from 40% to 55% [16]. A systematic review showed that treatment success is more likely when treatment is provided through the decentralized model [17]. New global guidelines recommend the use of decentralized and ambulatory models of care for MDR-TB treatment, but the recommendations are based on limited available evidence [18]. Our report from real-life programmatic experience further supports the global recommendations.

The TSR of different patient cohorts observed in this review ranged from 68% to 78%, which is higher than the global average of 60% [2]. Although progress has been observed in this trend over time, it is less than the WHO target set for the end of 2020 of ≥87% TSR [4]. Therefore, this progressive but suboptimal achievement may need further strengthening of programmatic implementation, expedited roll out of WHO recommended shorter and effective treatment regimens, and adoption of new WHO recommended diagnostic tools in the

country [19]. Different countries' treatment outcomes showed significant improvement with implementation of shorter treatment regimens [20, 21]. Data from other countries also suggest good treatment outcomes when intensive adverse event monitoring and management are put in place [22–24].

Ethiopia's ability to sustain a higher TSR while keeping up with multiple regimen transitions in an environment affected by multiple calamities is a lesson that other settings can learn from. Contributors to this success include strong technical and financial support from partners working on TB prevention and control, a strong NTP that facilitated the efficient use of available resources, rapid adoption of new recommendations and continuous capacity building of health care workers providing TB services [25]. The national technical working group and related technical subgroups worked to ensure that the services delivered were of the best quality possible. In line with WHO latest recommendation, the country updated its national guidelines in consultation with key stakeholders including the national clinical review committee and members of the national technical working group. The revised guidelines provide the latest guidance on patient management using the recommended regimens during different periods under programmatic conditions in Ethiopia.

Several programmatic actions contributed to effective, seamless transition to the new global recommendations. Some of these are decision made to fully adapt the new recommendations at different phases of the program scale-up; training of program managers and clinicians from TICs on the new policy changes; and stock analysis and supply planning.

Despite these successes, several challenges remain. Presumptive DR-TB patients are not routinely screened for DR-TB at all health facilities and are not referred from the community level. The use of DST for patients with extrapulmonary TB and childhood TB is suboptimal. The national treatment coverage rate is far below the national target. Treatment adherence support and tracking mechanisms for lost to follow-up patients are not well developed. Designing and evaluating interventions aimed at these challenges should be priorities for the next phase of the program. Specifically, supporting national data quality audit, and introduction of alternative private specimen transportation could address the data quality and access to X-pert testing, respectively.

The national TB program is also advocating for resource mobilization through developing Multisectoral Accountability Framework policy guide development to allow engagement of different sectors to finance TB services. Domestic resource mobilization is one area where partners and NTP are working together to allocate annual budget for the TB program.

An important limitation of this study is that we relied on secondary data review, and there were difficulties in getting complete information disaggregated by important variables like case detection by sex and treatment outcome by type of regimen. The types of documentation varied from year to year, leading to incompleteness of important variables.

In conclusion, this review showed an expansion and rapid decentralization of PMDT service with progressive improvement in DR-TB case detection, patient enrollment, TSR, and rapid uptake of new WHO recommended regimens. This is a result of strong local leadership combined with partner support. The remaining challenges should be addressed to further enhance the reach and quality of services. In addition, the NTP should consider an integrated DR-TB service with directly observed treatment and further decentralization of PMDT services to the community level.

## Supporting information

**S1 Fig. Diagramatic representation of the Ethiopian health tier system.** Available at file:/// C:/Users/ddare/AppData/Local/Temp/MicrosoftEdgeDownloads/b3197175-c020-40b1-b0e3-

0910f2ba0499/WHO-HIS-HSR-17.31-eng.pdf.
(JPG)

**S1 Dataset. Minimal data set.**
(XLSX)

## Acknowledgments

Our sincere and deepest gratitude goes to the USAID-supported Eliminate TB Project for its comprehensive and uninterrupted support of the TB program in Ethiopia.

We would like to thank all staff from the project and from federal and regional hospitals who were involved in the data collection and compilation.

## Author Contributions

**Conceptualization:** E. Tesema, D. G. Datiko, A. Gebreyohannes, Y. Molla, D. Jerene.

**Data curation:** E. Tesema, M. Biru, G. Gizatie, T. Bogale, M. Million, N. Hiruy, Z. G. Dememew.

**Formal analysis:** E. Tesema, M. Biru, Z. G. Dememew, D. Jerene.

**Funding acquisition:** D. G. Datiko.

**Investigation:** E. Tesema.

**Methodology:** E. Tesema, M. Biru, A. Gebreyohannes, Y. Molla, D. Jerene.

**Project administration:** E. Tesema, G. Gizatie, T. Bogale, M. Million, D. G. Datiko.

**Supervision:** E. Tesema, G. Gizatie, T. Bogale, M. Million, D. G. Datiko, D. Jerene.

**Validation:** E. Tesema, D. Jerene.

**Visualization:** E. Tesema, D. Jerene.

**Writing – original draft:** E. Tesema.

**Writing – review & editing:** E. Tesema, M. Biru, T. Leta, A. Kumsa, A. Liaulseged, D. G. Datiko, A. Gebreyohannes, Y. Molla, M. Mebnga, P. G. Suarez, Z. G. Dememew, D. Jerene.

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
