## [Decision Letter · Decision Letter 0]

10 Mar 2024

PONE-D-23-42213Drug-resistant tuberculosis care and treatment outcomes over the last 15 years in EthiopiaPLOS ONE

Dear Dr. Tesema,

Thank you for submitting your manuscript to PLOS ONE. After careful consideration, we feel that it has merit but does not fully meet PLOS ONE’s publication criteria as it currently stands. Therefore, we invite you to submit a revised version of the manuscript that addresses the points raised during the review process.

**Progress of the TB program in Ethiopia as discussed in this article provides a guide to programs in other countries. The guidance can be strengthened by discussing the results in light of the changing scenario over the period of assessment. Further analysis of data is warranted for this. Additionally, specific comments need to be individually addressed satisfactorily.  **

We look forward to receiving your revised manuscript.

Kind regards,

Yatin N. Dholakia, MD

Academic Editor

PLOS ONE

Journal Requirements:

Our sincere and deepest gratitude goes to the USAID-supported Eliminate TB Project for its comprehensive and uninterrupted support of the TB program in Ethiopia, including funding support.

We would like to thank all staff from the project and from federal and regional hospitals who were involved in the data collection and compilation.

5. Please amend the manuscript submission data (via Edit Submission) to include authors Dr. G. Gizatie, T. Bogale and M. Million.

6. Please upload a new copy of Figures 1, 2, 3 and 4 as the detail is not clear. Please follow the link for more information: https://blogs.plos.org/plos/2019/06/looking-good-tips-for-creating-your-plos-figures-graphics/" https://blogs.plos.org/plos/2019/06/looking-good-tips-for-creating-your-plos-figures-graphics/

Additional Editor Comments:

Progress of the TB program in Ethiopia as discussed in this article provides a guide to programs in other countries. The guidance can be strengthened by discussing the results in light of the changing scenario over the period of assessment. Further analysis of data is warranted for this.

Reviewers' comments:

Reviewer's Responses to Questions

**Comments to the Author**

1. Is the manuscript technically sound, and do the data support the conclusions?

Reviewer #1: Partly

Reviewer #2: Partly

2. Has the statistical analysis been performed appropriately and rigorously? 

Reviewer #1: N/A

Reviewer #2: No

3. Have the authors made all data underlying the findings in their manuscript fully available?

Reviewer #1: Yes

Reviewer #2: Yes

4. Is the manuscript presented in an intelligible fashion and written in standard English?

Reviewer #1: Yes

Reviewer #2: Yes

5. Review Comments to the Author

Reviewer #1: Thank you for the opportunity to review the manuscript Tesema et al. Drug-resistant tuberculosis care and treatment outcomes over the last 15 years in Ethiopia.

This paper is a timely contribution to an important topic. A couple of suggestions to strengthen the paper are noted below:

Overall

I think a very valuable piece of evidence but could be strengthened in a couple of key places to help readers who may be less familiar with the health service context of Ethiopia. Have made a couple of suggested.

Title

• Consider adding the study design to the title of the paper

Abstract

• Would benefit from more structure to the abstract. A couple of sentences on the background to the study would be beneficial.

• Could you say more about how many routine surveillance records was included in this analysis.

Main

• Does this study need permission from an ethics committee? My understanding is that the data are predominantly from published data sources or internal reports. None have been declared, though I commend the authors on seeking permission from one of the provincial health bureaus.

• Please clarify whether the routinely collected data used were patient-level data or aggregated summary data used for program management. Please clarify though which potential identifiers were removed to ensure anonymity – and who’s anonymity. Otherwise hard to assess.

• Pg3 ln 57 – please review this sentence as unclear

• Pg 3 ln 64 – suggest stating the burden of tuberculosis and DR-TB specifically in Ethiopia.

• Pg 4 ln 76 – would be good to know what the denominators are of these percentages.

• Were data used only from Oromia province. How generalisability are these findings to other provinces in Ethiopia?

• Pg 7, ln 131 – what does DRS stand for?

• Would suggest a paragraph summarising the context of DR TB care in Ethiopia, before launcing into the details. As currently names of hospitals are mentioned but may not be clear to the reader how they fit into wider service delivery

• Need more information on the statistical analysis done to produce estimates reported on page 10, ln 207.

• Can you explain a bit more why linked to the introduction of XPert MTB/RIF?

• Would be interesting to see data from Oromia versus other regions of Ethiopia – how are they different or similar?

• Pg 12, ln 235 – why were the utilisation of vans not as expected?

• Pg 12, ln 243 – what is HMIS

• These challenges mentioned focused on pastoralists/ people living in more rural regions – were there any important challenges to mention for the urban areas?

• Pg 13, ln 250 How was treatment access defined and assessed? Could improve this by saying … treatment access defined as XXX and assessed through our analysis of YYY data

• Pg 13 ln 251 – 1 what to 67 what?

• Pg 13 ln 261 – how was treatment success defined here?

• Pg 13, ln 267 what is TSR? Try to explain appreviations in each section of the paper before use and if used only once, rather spell it out

• Think that you could strengthen the paper by adding a paragraph describing Ethiopia’s economic situation ex. GDP per capita, proportion of budget spent on health and proportion of health spending from donor funding. Then discuss key health indicators – first overall then tuberculosis specific. Would then follow this with a descriptive paragraph on how the health system is generally organised and then how tuberculosis care and dr-TB care more generally fits into the health service.

Reviewer #2: The authors deserve appreciation for the commendable work in providing a snapshot of progress of TB programme in Ethiopia, specifically with DR-TB services. High treatment success rate among DR-TB patients would be exemplary for several other countries. However, it would be important to make the article more compact and focused by further analysis of the available data. Overall some major changes are required.

1. As of now the challenges and successes of the programme appear to be based on the assumptions of the authors and not coming out of analysis of the available data. While Table 1 presents the timelines of transition and, corresponding enrollment and treatment success rates, it does not demonstrate any link between them.

2. Other issues with Table 1 are - a) It is not easy to make out the annual enrollment while TSR is provided on annual basis, and b) It is not possible to figure out whether TSR is for the year or earlier cohort. This is specifically important to understand any changes to TSR with introduction of a different regimen.

3. It is also known that new tools, technologies and regimen have a gradual, phased introduction. This aspect is not clearly reflected in the manuscript -specifically proportion of population having access to these new developments.

4. As per the introduction, the aim of the manuscript is "assess Ethiopia’s experience in scaling up an ambulatory, decentralized model of care with a good treatment success rate (TSR) while managing multiple regimen transition processes and external shocks." However, the results do not come out very well answering this question, except some generic statements.

5. There are some challenges mentioned under "Challenges encountered during program implementation". However there is no further discussions on how these were specifically addressed or need to be addressed, such as low utilization of mobile vans, postal services, data quality.

6. Similar to #5 above, under the section "Data entry and analysis, one of the themes is "Resource limitation and lack of locally evidenced solutions". There is not much discussion on this aspect later in the manuscript.

7. Overall, it will also be good to include some specific leadership decisions, effective policy implementation process and planning methodology that led to success of the programme.

8. Minor comment - There is a lot of global information provided at the beginning that may not be relevant to the these of the manuscript. You could consider shortening it

6. PLOS authors have the option to publish the peer review history of their article (what does this mean?). If published, this will include your full peer review and any attached files.

Reviewer #1: No

Reviewer #2: **Yes: **Vineet Bhatia

---

## [Author Response · Author response to Decision Letter 0]

14 May 2024

24 April 2024

Subject: Point-by-point response to reviewers’ feedback

Dear Editor,

We would like to thank you and the reviewers for the constructive feedback on our manuscript. We addressed the comments point-by-point and made changes to the manuscript. We uploaded both the marked up and clear versions of the manuscript for your review. Our point-by-point responses are in bold to enable easy differentiation them from the reviewer comments. We have also amended the funding related information.

Looking forward to your favourable review,

Yours sincerely,

E.Tesema, primary author

PONE-D-23-42213

Drug-resistant tuberculosis care and treatment outcomes over the last 15 years in Ethiopia

PLOS ONE

Dear Dr. Tesema,

Thank you for submitting your manuscript to PLOS ONE. After careful consideration, we feel that it has merit but does not fully meet PLOS ONE’s publication criteria as it currently stands. Therefore, we invite you to submit a revised version of the manuscript that addresses the points raised during the review process.

Progress of the TB program in Ethiopia as discussed in this article provides a guide to programs in other countries. The guidance can be strengthened by discussing the results in light of the changing scenario over the period of assessment. Further analysis of data is warranted for this. Additionally, specific comments need to be individually addressed satisfactorily. 

Response: Thank you for the compliment

Thank you for stating the following in the Acknowledgments Section of your manuscript: 

Our sincere and deepest gratitude goes to the USAID-supported Eliminate TB Project for its comprehensive and uninterrupted support of the TB program in Ethiopia, including funding support.

We would like to thank all staff from the project and from federal and regional hospitals who were involved in the data collection and compilation.

Response: This is well noted. We have removed the funding related information.

Amended Statement 

The Global Health Bureau, Office of Health, Infectious Disease, USAID financially supported the study through USAID Eliminate TB Project under the terms of Agreement No. 72066320CA00009. The authors’ views expressed in this publication do not necessarily reflect the views of USAID or the US Government. 

We note that your Data Availability Statement is currently as follows: All relevant data are within the manuscript and its Supporting Information files.

Response: We uploaded the minimal data set as supporting information.

PLOS requires an ORCID iD for the corresponding author in Editorial Manager on papers submitted after December 6th, 2016. Please ensure that you have an ORCID iD and that it is validated in Editorial Manager. To do this, go to ‘Update my Information’ (in the upper left-hand corner of the main menu), and click on the Fetch/Validate link next to the ORCID field. This will take you to the ORCID site and allow you to create a new iD or authenticate a pre-existing iD in Editorial Manager. Please see the following video for instructions on linking an ORCID iD to your Editorial Manager account: https://www.youtube.com/watch?v=_xcclfuvtxQ

Response: ORCID IDs are updated

Please amend the manuscript submission data (via Edit Submission) to include authors Dr. G. Gizatie, T. Bogale and M. Million.

Response: Manuscript submission data amended and missing authors are included.

Please upload a new copy of Figures 1, 2, 3 and 4 as the detail is not clear. Please follow the link for more information: https://blogs.plos.org/plos/2019/06/looking-good-tips-for-creating-your-plos-figures-graphics/" https://blogs.plos.org/plos/2019/06/looking-good-tips-for-creating-your-plos-figures-graphics/

Response: Clearer copies of figures are uploaded and figure 1 is modified to incorporate the information from Table 1.

Please include captions for your Supporting Information files at the end of your manuscript, and update any in-text citations to match accordingly. Please see our Supporting Information guidelines for more information: http://journals.plos.org/plosone/s/supporting-information. 

Response: Captions for Supporting Information added at the end of the manuscript

Additional Editor Comments:

Progress of the TB program in Ethiopia as discussed in this article provides a guide to programs in other countries. The guidance can be strengthened by discussing the results in light of the changing scenario over the period of assessment. Further analysis of data is warranted for this.

Response: This is well noted, and we updated the data analysis for the current scenario

Responses to reviewer 1:

Reviewer #1: Thank you for the opportunity to review the manuscript Tesema et al. Drug-resistant tuberculosis care and treatment outcomes over the last 15 years in Ethiopia.

This paper is a timely contribution to an important topic. A couple of suggestions to strengthen the paper are noted below:

Overall

I think a very valuable piece of evidence but could be strengthened in a couple of key places to help readers who may be less familiar with the health service context of Ethiopia. Have made a couple of suggested.

Response: We appreciate the compliments

Title

• Consider adding the study design to the title of the paper

Response: added as suggested

Abstract

• Would benefit from more structure to the abstract. A couple of sentences on the background to the study would be beneficial.

• Could you say more about how many routine surveillance records was included in this analysis.

Response: Comment noted and abstract amended

Main

• Does this study need permission from an ethics committee? My understanding is that the data are predominantly from published data sources or internal reports. None have been declared, though I commend the authors on seeking permission from one of the provincial health bureaus.

• Please clarify whether the routinely collected data used were patient-level data or aggregated summary data used for program management. Please clarify though which potential identifiers were removed to ensure anonymity – and who’s anonymity. Otherwise hard to assess.

Response: We used aggregated summary data used for program management. No patient-level data was collected, nor did we do any interviews with individuals. In principle, no ethics review is needed for this kind of data. However, we wanted to emphasize that the project under which this study was conducted had secured blanket ethical coverage for any potential inclusion of individual-level data based on a generic protocol. 

• Pg3 ln 57 – please review this sentence as unclear

Response: Sentence amended

• Pg 3 ln 64 – suggest stating the burden of tuberculosis and DR-TB specifically in Ethiopia.

Response: Ethiopia specific burden added, and the background section substantially updated.

• Pg 4 ln 76 – would be good to know what the denominators are of these percentages.

• Were data used only from Oromia province. How generalisability are these findings to other provinces in Ethiopia?

Response: We provided further details about the denominators and numerators in the results section. Data is for the whole country. Oromia and Amhara are mentioned because this is where we had specific ethical approval for individual-level data analysis which we explained above. 

• Pg 7, ln 131 – what does DRS stand for?

Response: DRS stands for Drug Resistance Survey—its full form was available in the background section.

• Would suggest a paragraph summarizing the context of DR TB care in Ethiopia, before launching into the details. As currently names of hospitals are mentioned but may not be clear to the reader how they fit into wider service delivery

Response: This comment is well noted and we added a paragraph under the setting: Ethiopia is a low-income country. The country’s population, which is predominantly rural, is estimated to exceed 123 million and the average life expectancy is 65 years. Table 1 summarizes key socio-demographic and health indicators for Ethiopia. The country has a three-tiered pyramidal health system care system with primary health care facilities forming the broad base, secondary health care facilities in the middle, and tertiary care at the apex (See Figure as supplemental information). Ethiopia follows a decentralized service delivery model as much as possible with the bulk of the service provided at primary care level. For DR-TB care, the country followed the healthcare tier system in setting up the services. Treatment Initiating Centers (TICs), where patients initiate their treatment and temporarily admitted if needed were initially set up at tertiary level but later much of the TIC role moved to secondary level. Treatment follow-up centers (TFCs) are facilities where patients receive their daily treatment after they are transferred from the TICs. Each TIC has a designated number of TFC within their catchment areas, defined by administrative regions and zones. Detailed description of the roles and responsibilities of TICs and TFCs is provided in reference 7.

• Need more information on the statistical analysis done to produce estimates reported on page 10, ln 207.

Response: We thank the reviewer for this observation. After through discussion among authors, we decided to remove the correlation analysis because of potentials for huge compounding.

• Can you explain a bit more why linked to the introduction of XPert MTB/RIF?

Response: The introduction of Xpert MTB/RIF was linked to more Rifampicin resistant cases. Prior to the Xpert introduction, DR-TB diagnosis was done through culture and DST, which takes 6-8 weeks, when available. 

• Would be interesting to see data from Oromia versus other regions of Ethiopia – how are they different or similar?

Response: Our goal was to show trends in national data. We do not expect any unique trends in Oromia. Oromia was mentioned in connection with the ethics review and approval as explained earlier. 

• Pg 12, ln 235 – why were the utilisation of vans not as expected?

Response: This appeared to be linked to lack of experience with the use of vans. We added a sentence about this. 

• Pg 12, ln 243 – what is HMIS

Response: HMIS stands for Health Management Information System. This was explained under Data collection and procedures.

• These challenges mentioned focused on pastoralists/ people living in more rural regions – were there any important challenges to mention for the urban areas?

Response: We did not identify challenges worth reporting.

• Pg 13, ln 250 How was treatment access defined and assessed? Could improve this by saying … treatment access defined as XXX and assessed through our analysis of YYY data

Response: This was defined by the number of TICs which increased nearly 70-fold. The text is slightly amended. 

• Pg 13 ln 251 – 1 what to 67 what?

Response: TICs, this is in the sentence itself, but we made minor modifications to make it clearer.

• Pg 13 ln 261 – how was treatment success defined here?

Response: Treatment success is defined according to the national/global guidelines, i.e, treatment completed plus cured divided by total cohort evaluated. 

• Pg 13, ln 267 what is TSR? Try to explain appreviations in each section of the paper before use and if used only once, rather spell it out

Response: TSR stands for treatment success rate. All abbreviations were spelled out in the preceding sections.

• Think that you could strengthen the paper by adding a paragraph describing Ethiopia’s economic situation ex. GDP per capita, proportion of budget spent on health and proportion of health spending from donor funding. Then discuss key health indicators – first overall then tuberculosis specific. Would then follow this with a descriptive paragraph on how the health system is generally organised and then how tuberculosis care and dr-TB care more generally fits into the health service.

Response: This comment is well noted and we made substantial improvement to the flow of the background information as well as description of the study setting. A new table summarizing the socio-economic and health indicators is added and we included supplemental figure describing the Ethiopian healthcare system.

Reviewer #2: The authors deserve appreciation for the commendable work in providing a snapshot of progress of TB programme in Ethiopia, specifically with DR-TB services. High treatment success rate among DR-TB patients would be exemplary for several other countries. However, it would be important to make the article more compact and focused by further analysis of the available data. Overall some major changes are required.

Response: We appreciate the reviewer’s constructive feedback and made changes as suggested.

1. As of now the challenges and successes of the programme appear to be based on the assumptions of the authors and not coming out of analysis of the available data. While Table 1 presents the timelines of transition and, corresponding enrollment and treatment success rates, it does not demonstrate any link between them.

Response: The challenges and successes are based on qualitative dat

---

## [Decision Letter · Decision Letter 1]

11 Jun 2024

Drug-resistant tuberculosis care and treatment outcomes over the last 15 years in Ethiopia: results from a mixed-method review of trends

PONE-D-23-42213R1

Dear Dr. Emawayish Tesema

We’re pleased to inform you that your manuscript has been judged scientifically suitable for publication and will be formally accepted for publication once it meets all outstanding technical requirements.

Kind regards,

Yatin N. Dholakia, MD

Academic Editor

PLOS ONE

Additional Editor Comments (optional):

Progress of the TB program in Ethiopia as discussed in this article provides a guide to programs in other countries.

Reviewers' comments:

Reviewer's Responses to Questions

**Comments to the Author**

1. If the authors have adequately addressed your comments raised in a previous round of review and you feel that this manuscript is now acceptable for publication, you may indicate that here to bypass the “Comments to the Author” section, enter your conflict of interest statement in the “Confidential to Editor” section, and submit your "Accept" recommendation.

Reviewer #2: All comments have been addressed

2. Is the manuscript technically sound, and do the data support the conclusions?

Reviewer #2: Yes

3. Has the statistical analysis been performed appropriately and rigorously? 

Reviewer #2: Yes

4. Have the authors made all data underlying the findings in their manuscript fully available?

Reviewer #2: Yes

5. Is the manuscript presented in an intelligible fashion and written in standard English?

Reviewer #2: Yes

6. Review Comments to the Author

Reviewer #2: (No Response)

7. PLOS authors have the option to publish the peer review history of their article (what does this mean?). If published, this will include your full peer review and any attached files.

Reviewer #2: **Yes: **Vineet Bhatia

---

## [Editor Report · Acceptance letter]

27 Jun 2024

PONE-D-23-42213R1 

PLOS ONE

Dear Dr. Tesema, 

I'm pleased to inform you that your manuscript has been deemed suitable for publication in PLOS ONE. Congratulations! Your manuscript is now being handed over to our production team.

Kind regards, 

on behalf of

Dr. Yatin N. Dholakia 

Academic Editor

PLOS ONE